# Prediction of Strain in Embedded Rebars for RC Member, Application of Hybrid Learning Approach

Ali Mirzazade [1,*], Cosmin Popescu [1,2] and Björn Täljsten [1]

1   Structural and Fire Engineering, Department of Civil, Environmental and Natural Resources Engineering, Luleå University of Technology, 97187 Luleå, Sweden
2   SINTEF Narvik AS, 8517 Narvik, Norway
*   Correspondence: ali.mirzazade@ltu.se; Tel.: +46-(0)920-493613 or +46-(0)-73-059-68-45

**Abstract:** The aim of this study was to find strains in embedded reinforcement by monitoring surface deformations. Compared with analytical methods, application of the machine learning regression technique imparts a noteworthy reduction in modeling complexity caused by the tension stiffening effect. The present research aimed to achieve a hybrid learning approach for non-contact prediction of embedded strains based on surface deformations monitored by digital image correlation (DIC). However, due to the small training dataset collected by the installed strain gauges, the input dataset was enriched by a semi-empirical equation proposed in a previous study. Therefore, the present study discussed (i) instrumentation by strain gauge and DIC as well as data acquisition and post-processing of the data, accounting for strain gradients on the concrete surface and embedded reinforcement; (ii) input dataset generation for training machine learning regression models approaching hybrid learning; (iii) data regression to predict strains in embedded reinforcement based on monitored surface deformations; and (iv) the results, validation, and post-processing responses to make the method more robust. Finally, the developed model was evaluated through numerous statistical performance measures. The results showed that the proposed method can reasonably predict strain in embedded reinforcement, providing an innovative type of sensing application with highly improved performance.

**Keywords:** machine learning; hybrid learning; digital image correlation; neural network; Gaussian process regression; decision tree; ensemble model; strain gauge; reinforced concrete; strain





## 1. Introduction

Concrete cracking is usually an indicator of RC structure performance; therefore, crack mapping and assessment in RC members is an important step in concrete infrastructure inspection. In addition to cracks, reinforcement strain, which is usually accompanied by bond–slip and concrete cracking, has particular importance in evaluating a structure's status and safety. Observed strains that are close to the yielding strain of the reinforcement indicate structure overloading or that a failure is about to occur.

Electrical resistance strain gauges (SGs) and fiber optic sensors (FOSs) are commonly used to measure reinforcement strain, but they need to be attached to the reinforcement for monitoring. In existing structures, this usually requires removing the concrete cover and exposing the reinforcement. In this case, sensors are referred to as being post-installed. However, for new structures, sensors can be pre-installed on the reinforcement in the construction phase.

Regardless of the installation method, i.e., pre-installed or post-installed, SGs have a number of drawbacks; for instance, measurements recorded by SGs represent local strains averaged over the gauge's length, while the highest reinforcement strain is at the location of cracks. These cracks are not always immediately obvious, and it is not feasible to predict the exact location of cracks in a structure before they occur. This means cracks will appear at scattered locations depending on locally weak sections.

However, optical methods such as the digital image correlation (DIC) system overcome the limitation of only providing local measurements and allow us to monitor the surface in

a two- or three-dimensional system. According to Hoult et al. [1], DIC has the potential to be a new alternative to traditional technologies used to assess RC structures. DIC is a technique that was initiated in the 1980s [2] and has been used in several recent studies, such as embedded rebar assessment using surface deformation [3–5].

## 2. Prior Work and Research Need

Advancements in optical technique algorithms have led engineers to achieve rapid implementation and convenience in structural inspection. Therefore, there is increasing interest in image-based inspection for structural health monitoring (SHM). However, one of the challenges associated with assessing existing infrastructure is correlating externally measured parameters, such as surface strains, with in-depth parameters, such as reinforcement strains, which depends on a number of independent input variables. The correlation between surface deformation and strain in embedded reinforcement has been studied in some research [3,4,6,7], showing the feasibility of embedded strain prediction using surface measurements. Therefore, the authors studied the idea of strain measurement in embedded reinforcement by generating a semi-empirical equation in a previously published article [5]. As a result, good correlation was observed between the strain in embedded reinforcement and surface deformation, which were monitored by installed strain gauges and a digital image correlation system, respectively. However, the proposed method still needs improvement because (1) the data collected by existing pre-installed strain gauges were limited (in few locations), and (2) the proposed equation had many variables, which made computation difficult. Here, the application of machine learning aimed to find a regression between all the collected independent input variables and monitored strains in embedded reinforcement, without the complexities that exist in the application of empirical equations.

### 2.1. Research Description and Significance

In a previous study by the authors [5], strain gauges were installed in only a few locations through the embedded reinforcement. Therefore, there were not enough available data to meticulously train the ML regression model. Thus, the use of hybrid learning was proposed in this study to feed the model with both experimental data and synthetic data generated by a previously published semi-empirical equation [5]. To this end, data collected from installed strain gauges usually located between the appearing cracks with no bond–slip were combined with additional synthetic data for the crack zones and around the debonding/slip zones. The theory behind this equation was based on the tension stiffening effect, which can simulate the debonding and bond–slip zone around the cracks. Then, machine learning (ML) regression models were generated and trained, approaching a hybrid learning method. The developed models were evaluated through numerous statistical performance measures to find the most efficient trained model. Finally, if we were satisfied with the performance of the trained model, it was deployed on a new sample with no previously observed data. Otherwise, the input dataset needed to be checked and prepared for training by the new optimized regression models. Figure 1 shows the proposed workflow, which will be defined in detail in this paper.

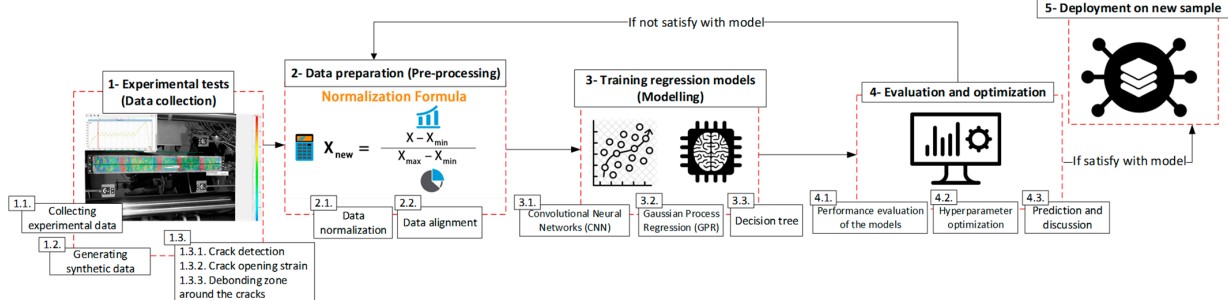

**Figure 1.** The schematic workflow of the proposed hybrid learning approach.

This study aimed to deploy innovative and intelligent instruments for strain estimation along embedded rebar for SHM purposes in civil engineering structures.

## 2.2. Theoretical Approach in Tensile Behavior of RC Members

The constitutive law adopted to simulate the tensile behavior of the RC member is called CEB Model [8], which was developed for non-linear plastic behavior resulting from the tension stiffening effect. The tension stiffening effect is described as the contribution of the surrounding concrete to the tensile stiffness of the reinforcing bars. This stiffness is provided by uncracked and bond–slip zones generated by the strain localization process [9]. The conceptual model of RC members subjected to monotonic tensile loading, shown in Figure 2, presents different behaviors of RC members, including (1) uncracked concrete with elastic behavior, (2) the cracking phase, and (3) stabilized cracking, while there is still debonding development around the cracks.

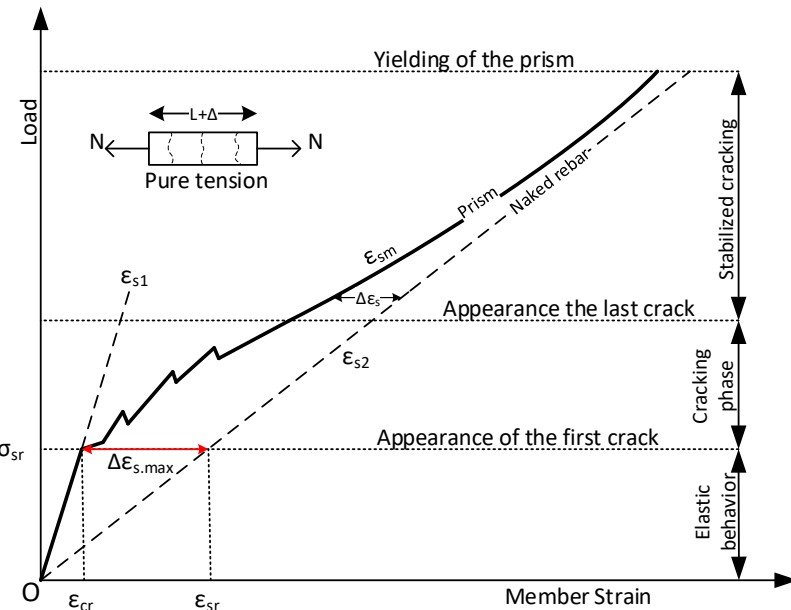

**Figure 2.** Conceptual model of tensile behavior, including identification of the most relevant stages. Discussed by ref. [10] in detail.

The first step in elastic behavior, hardening of the zone until a planar crack forms, occurs just before the maximum principal stress ($\sigma_{sr}$) is reached. It can be calculated by Equation (1) [11]:

$$\sigma_{sr} = \frac{1 + \eta\rho}{\rho} f_{ct},\tag{1}$$

where the reinforcement ratio is $\rho = {A_s}/{A_c}$, $\eta = {E_s}/{E_c}$, and $f_{ct}$ is the concrete tensile strength. By increasing the tensile stress, bond–slip and crack propagation form at random locations according to locally weak sections. Crack planes are perpendicular to the direction of pure tension and may slide if the direction of principal stress changes.

Crack propagation takes place in the cracking phase with observed crack opening strain, $\varepsilon_{cr}$, on detected cracks. Then, after a complete cracking process zone, no new cracks appear in the stabilized cracking phase, and only the opening strain in existing cracks increases. Accordingly, bonding deterioration around the cracks develops, namely the debonding length, $L_{debonding}$, given by Equation (4). The change in crack width $\Delta w$ is calculated as the total crack opening displacement within the debonding length:

$$\Delta w = \varepsilon_{cr} L_{debonding},\tag{2}$$

$$\sigma = E_s \varepsilon_{cr}, \tag{3}$$

$$L_{debonding} = \frac{E_s \Delta w}{\sigma}, \tag{4}$$

where $\varepsilon_{cr}$ is the crack opening strain, while a concrete crack is initiated if the strain exceeds 0.010% to 0.012% [12]. The axial stiffness of the RC member proceeds toward the naked rebar via crack propagation, developing debonding zones and decreasing concrete collaboration in transferring tensile stress, as shown in Figure 2. Therefore, this behavior, called the tension stiffening effect, can be defined as a non-linear model. As illustrated in Figure 3, the contribution of concrete to tension is a non-linear function of the concrete's strength, geometry, reinforcement ratio, bond properties of the reinforcement, and the modular ratio [13]. '

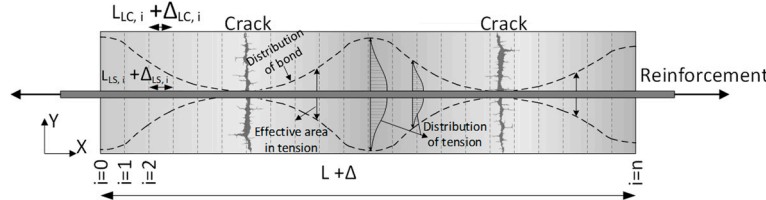

**Figure 3.** Conceptual figure of the effective area in tension.

Accordingly, a semi-empirical equation was proposed by the authors in a previous study [5], which concurred with the non-linearity resulting from the tension stiffening effect. Therefore, the local strain ($\varepsilon_{LS, i}$) in embedded rebar can be estimated by the local surface strain ($\varepsilon_{LC, i}$) in the cracking and stabilized cracking phases, as given by Equation (5):

$$\sum_{i=0}^{n} \varepsilon_{LS, i} = \left( \frac{f_{ct}(1 + \eta\rho)}{\rho\sigma_s} \right)^{\alpha} \sum_{i=0}^{n} \varepsilon_{LC, i} + \left[ 1 - \left( \frac{f_{ct}(1 + \eta\rho)}{\rho\sigma_s} \right)^{\alpha} \right] \frac{\sigma_s}{E_s}, \tag{5}$$

In the plastic behavior zone ($if\ \sigma_s > \sigma_{sr}$), the local strain ($i = n$) in the embedded rebar is $\varepsilon_{LS, i} = \frac{\Delta_{LS, i}}{L_{LS, i}}$ and the local strain ($i = n$) on the surface of the concrete is $\varepsilon_{LC, i} = \frac{\Delta_{LC, i}}{L_{LC, i}}$.

$$\sigma_s = \frac{E_s \left( \varepsilon - \frac{\sigma_{sr}}{E_{RC}} \right)^2}{\left( \varepsilon - \underbrace{\varepsilon_{cr}}_{\left( \frac{\sigma_{sr}}{E_{RC}} \right)} \right) + \left( \frac{\sigma_{sr}}{E_s} - \underbrace{\varepsilon_{cr}}_{\left( \frac{\sigma_{sr}}{E_{RC}} \right)} \right)} + \sigma_{sr}, \tag{6}$$

where $\sigma_s$ is the average stress level in the intended part, formulated by Khalfallah and Guerdouh [11]; $\varepsilon$ is the average strain along the surface, which is patterned and monitored by DIC; $E_{RC}$ is the elastic modulus of the reinforced concrete; and $\alpha$ is a constant value based on the concrete material and geometry presented earlier. This $\alpha$ value was experimentally calculated for two types of concrete and geometry [5].

Since Equation (5) has many variables, which makes it complex to apply in the field, ML regression models were trained using two datasets obtained by (1) strain gauges in bonding zones and (2) a semi-empirical equation in slip/debonding zones combined with the SG dataset, thus approaching hybrid learning.

## 2.3. Machine Learning and Approach of Hybrid Learning

In recent years, machine learning models have been employed as a tool that uses mathematical formulations to construct a brain-type learning system operation based on the relationships that exist between observed inputs and target outputs. In other words,

regression analysis refers to the method of studying the correlations between independent input variables and dependent target values.

One of the challenges in this study was the small amount of data collected by the strain gauges installed in limited positions in order to train the ML regression models. Accordingly, implementing state-of-the-art synthetic data generators can provide us with a bigger dataset meant to develop larger and improved training datasets without depreciating the original application. This synthetic dataset was provided using the semi-empirical equation proposed by Mirzazade et al. [5] for zones in the debonding length around the appearing cracks. Then, by learning the complex distribution of a dataset, approaching hybrid learning, the ML regression models were trained to predict strain in embedded reinforcement.

## 3. Operational Principle and Experimental Set-Up for Data Accumulation

The experimental tests were carried out to meticulously collect part of the input training datasets in the bonding zone. The main objective of this section is to describe the details of the operational principle of the experiment and collection of data to implement the ML regression models.

### 3.1. Material and Experimental Procedures

The prepared specimens were casted using two different concrete properties, Normal Concrete (NC) and High-Performance Concrete (HPC), in two different cross-sectional areas: $100 \times 100$ mm$^2$ and $150 \times 150$ mm$^2$. All the recipes for the four prepared batches in both NC and HPC are presented in Table 1.

**Table 1.** Normal concrete and HPC recipes.

| Concrete Recipes for Batches 1–3 of NC | | | HPC Recipe | |
|---|---|---|---|---|
| Batch | 1 | 2,3 | Batch | 4 |
| Volume [L] | 45 | 30 | Volume [L] | 32 |
| Cement [kg] | 17.10 | 11.40 | Cement [kg] | 32 |
| Water [kg] | 8.725 | 5.817 | Silica fume [kg] | 6.4 |
| Aggregate 0–4 mm [kg] | 58.05 | 38.70 | Quartz [kg] | 9.6 |
| Aggregate 4–8 mm [kg] | 19.35 | 12.90 | Water [kg] | 7.36 |
| Filler [kg] | 1.800 | 1.200 | Superplasticizer [kg] | 0.48 |
| Superplasticizer [kg] | 0.1125 | 0.075 | Sand Type I [1] [kg] | 11.2 |
| | | | Sand Type II [1] [kg] | 11.2 |

[1] 'Sand Type I' and 'Sand Type II' are two different sand types; type I is finer.

In this test, the reinforcement was a hot rolled ribbed 16 mm bar, type B500B, with 500 MPa yield strength (Re), 1.08 tensile/yield strength ratio (Rm/Re), and 200 GPa Young's modulus. The RC members were thereafter cast and left to cure for 28 days. The compressive test was carried out on prepared test cubes after 28 days of curing, and the obtained mechanical characteristics are shown in Table 2.

**Table 2.** Mechanical properties of normal concrete and HPC.

| | W/C | Compressive Strength [1] [MPa] ($f_{ck}$) | Tensile Strength [2] [MPa] ($f_{ct}$) | Young Modulus [2] [GPa] ($E_{cm}$) |
|---|---|---|---|---|
| Normal Concrete | 0.5 | 50.63 | 2.0 | 35 |
| HPC | 0.23 | 115.20 | 4.5 | 70 |

[1] Measured at 28 days on cubes according to standard EN 12390-1. [2] Measured at 28 days on cubes according to standard EN 1992-1-1: 2005.

### 3.2. Instrumentation
#### 3.2.1. Strain Gauges

Initially, the rebars were cut to a length of 1500 mm. The tension concrete prism had an 800 mm long reinforcement in the middle of the cross-section while the bars

extended outside the concrete prism by 350 mm on each side, as shown in Figure 4. Eight prepared specimens were equipped with eight embedded SGs and two other specimens were equipped with 15 SGs. In the rebars with eight SGs, the first one was located at approximately 50 mm from the edge of the concrete and the distance between the gauges was approximately 100 mm. In the rebars with 15 SGs, the distance between them was approximately 50 mm, as shown in Figure 4. All attached strain gauges were connected to a computer in the laboratory and recorded at a rate of 10 Hz by a data acquisition system.

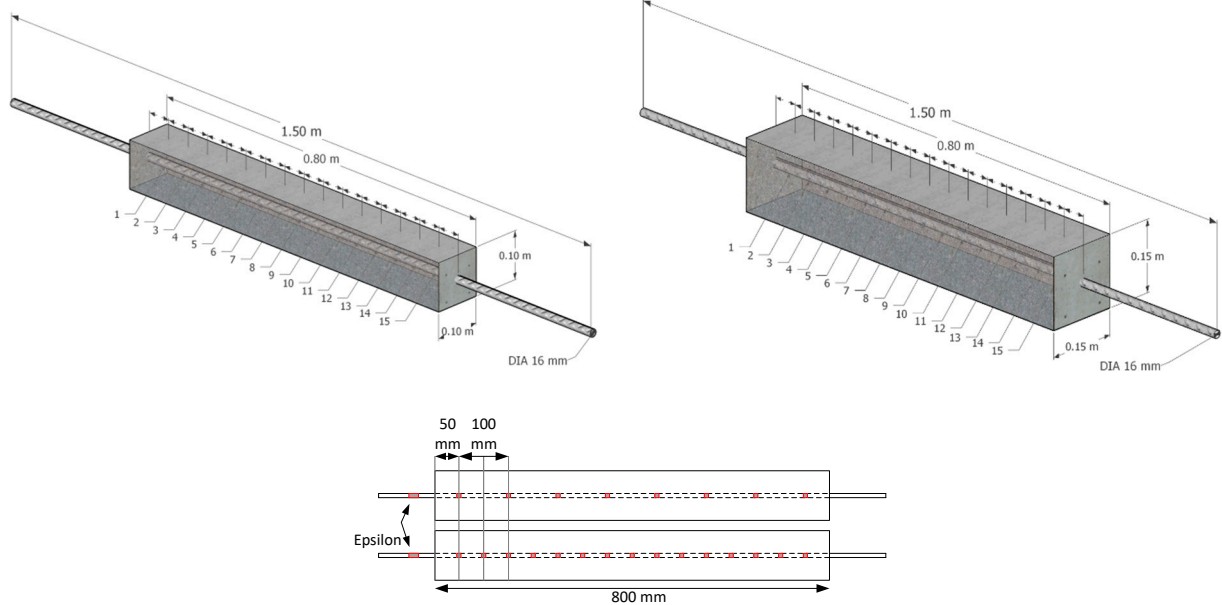

**Figure 4.** Geometry of specimens with locations of installed strain gauges.

To install the SGs, a belt grinder was used to grind the areas where the SGs were located, and sandpaper was used to further smooth the surface. The strain gauge was removed from the protective plastic and normal tape was applied to the top of the strain gauge. The tape was used to apply the gauge straight along the rebar and lift one side of the SG. Then, adhesive (Rapid Adhesive Z70 by HBM) was applied to the strain gauge and connecting cables, as shown in Figure 5.

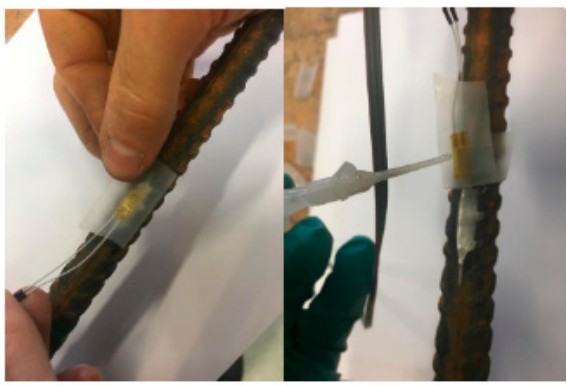

**Figure 5.** Installing SGs on rebars.

Overall, 10 specimens were prepared in three different groups based on concrete type and cross-sectional area, comprising: (1) NC—$100 \times 100$ mm$^2$, (2) NC—$150 \times 150$ mm$^2$, (3) HPC—$100 \times 100$ mm$^2$. The specimens, with the exact location of the installed SGs from the edge of the concrete (0–800 mm), are summarized in Table 3. After casting the

specimens and testing the installed SGs, there was no response from the installed SGs on 5N100, meaning that no data were retrieved from 5N100.

**Table 3.** Tested specimens.

| ID | Group | Dimension [mm²] | No. SGs | Type | SG Locations [mm] | | | | | | | | | | | | | | |
|---|---|---|---|---|---|---|---|---|---|---|---|---|---|---|---|---|---|---|---|
| | | | | | 1 | 2 | 3 | 4 | 5 | 6 | 7 | 8 | 9 | 10 | 11 | 12 | 13 | 14 | 15 |
| 1N150 | 2 | 150 × 150 | 15 | NC | 51 | 100 | 151 | 200 | 253 | 301 | 352 | 402 | 452 | 502 | 550 | 605 | 655 | 704 | 754 |
| 2N100 | 1 | 100 × 100 | 15 | NC | 46 | 97 | 146 | 195 | 246 | 299 | 349 | 397 | 447 | 499 | 548 | 598 | 647 | 698 | 748 |
| 3N150 | 2 | 150 × 150 | 8 | NC | 50 | - | 153 | - | 253 | - | 353 | - | 454 | - | 551 | - | 656 | - | 756 |
| 4N150 | 2 | 150 × 150 | 8 | NC | 49 | - | 149 | - | 248 | - | 349 | - | 449 | - | 549 | - | 648 | - | 747 |
| 5N100 | 1 | 100 × 100 | 8 | NC | - | - | - | - | - | - | - | - | - | - | - | - | - | - | - |
| 6N100 | 1 | 100 × 100 | 8 | NC | 43 | - | 145 | - | 247 | - | 348 | - | 449 | - | 549 | - | 648 | - | 747 |
| 7N100 | 1 | 100 × 100 | 8 | NC | 53 | - | 154 | - | 253 | - | 353 | - | 452 | - | 552 | - | 651 | - | 752 |
| 8U100 | 3 | 100 × 100 | 8 | HPC | 51 | - | 151 | - | 251 | - | 351 | - | 450 | - | 550 | - | 652 | - | 752 |
| 9U100 | 3 | 100 × 100 | 8 | HPC | 48 | - | 148 | - | 248 | - | 349 | - | 450 | - | 550 | - | 650 | - | 750 |
| 10U100 | 3 | 100 × 100 | 8 | HPC | 47 | - | 149 | - | 249 | - | 350 | - | 448 | - | 550 | - | 649 | - | 747 |

### 3.2.2. Surface Preparation (DIC)

Digital image correlation (DIC) is a non-contact measurement technique that uses digital images to obtain surface deformations. For this aim, the surface of the specimen must be prepared. Holes on the surfaces can disturb image correlation, be recognized as speckles, and may have an impact on the result. To prevent this, holes on the surface were filled using wall putty. The surface was then painted using matte white spray paint. Once the white spray paint had dried, a speckle pattern was painted on the surface using black spray paint. According to Reu [2], each speckle should have a diameter of at least 3 pixels. The other important parameter is coverage, which is the ratio of black-to-white pixels in the entire pattern and should be between 40–70% according to Mazzoleni [14], but ideally around 50% according to Reu [2]. For this test, the speckle diameter in the pattern was 7.14 pixels and the coverage was 59.0%, which was recognized as the most suitable pattern for this test. The schematic of the experimental setup is shown in Figure 6. The used DIC system captured images in 2448 × 2050 pixel resolution and the measured area focused on the specimen in 1005 × 880 mm in the current set up.

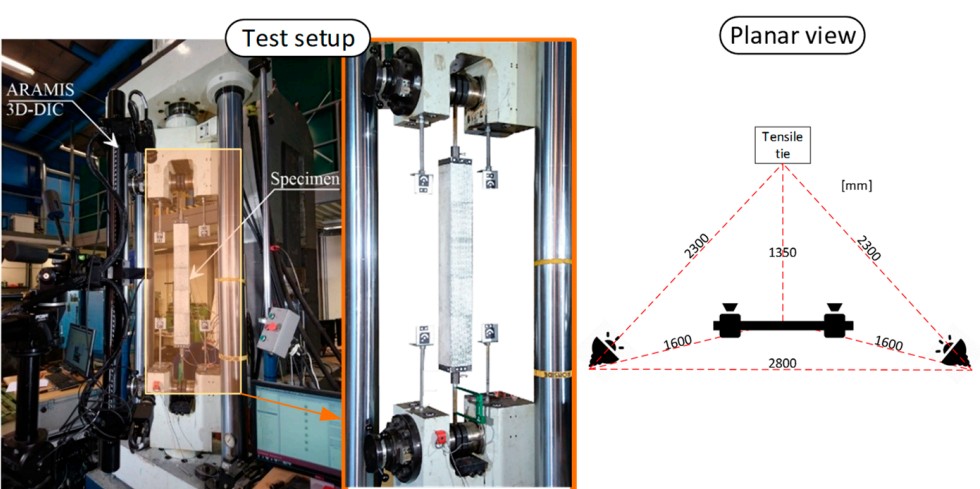

**Figure 6.** Test setup by DIC system [images taken by Jenny Lindmark, Master thesis, LTU].

### 3.3. Test Set-Up and Obtained Results

### 3.3.1. Uniaxial Tension Test

The prepared specimens were tested under pure tension using a 600 kN testing machine, where the reinforcement on both sides of the specimen was secured. Specimens were aligned in the machine; thus, the surface of the specimen was perpendicularly aligned

to the DIC system. To enable the software to measure the total displacement of the machine, the component "Stroke" was created defining the distance between the plates attached to the bottom and top grip of the machine, as shown in Figure 7. On the plates of the testing machine connected to the fixed reinforcement, two components were created, "ReinfTop" and "ReinfBot", and the component "AvgBarStrain" was defined to visualize the rebar average strain in the DIC system, as shown in Figure 7.

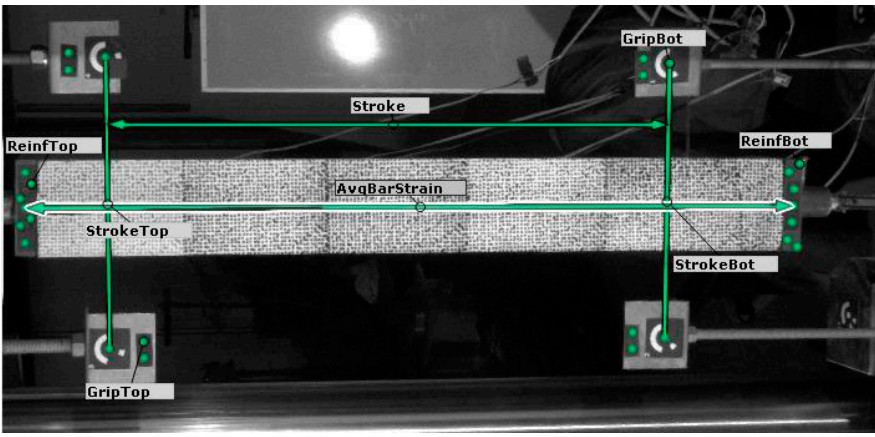

**Figure 7.** Installed specimen on uniaxial testing machine with defined components.

The specimens were tested using cyclic load sequences with an amplitude of about 200 μm/m in each. Therefore, in a rebar with a diameter of 16 mm, a load variation of 10 kN was calculated by combining Hooke's law and Navier's formula. Then, three different loading patterns for each group of specimens were applied, as shown in Figure 8.

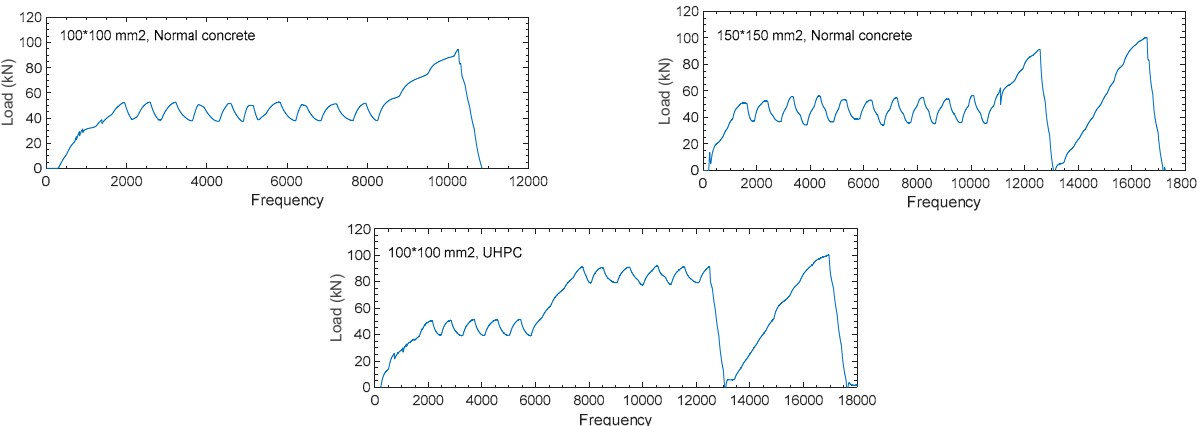

**Figure 8.** Load sequences for: (1) 100 × 100 mm$^2$—Normal concrete, (2) 150 × 150 mm$^2$—Normal concrete, and (3) 100 × 100 mm$^2$—HPC.

### 3.3.2. Experimental Results

The testing machine applied monotonic tensile loading according to the designed loading patterns. Meanwhile, strains on the embedded reinforcement and surface deformations were monitored by the installed strain gauges and DIC system, respectively. As an example, Figure 9 illustrates the surface strain and displacement monitored by DIC on sample 2N100 under 94.14 kN tensile load.

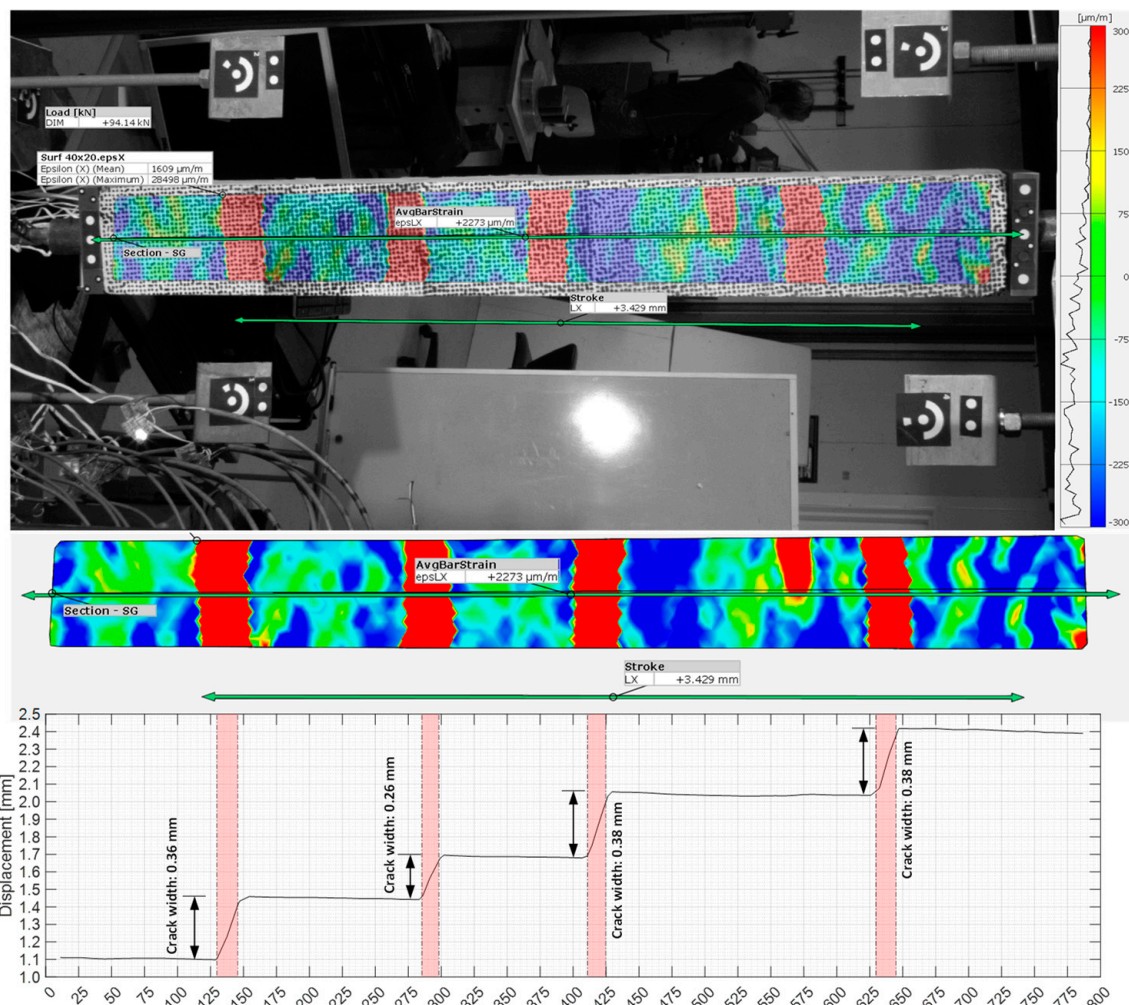

**Figure 9.** Monitored surface strain and displacement on sample 2N100 under 94.14 kN tensile load.

Appearing cracks can be detected as a jump in the surface displacement [15]. Figure 10 presents the location of four propagating cracks on 2N100, in order of propagation, under six different loading levels. Accordingly, Table 4 summarizes the loading levels reached by the testing machine at the time that cracks formed for all specimens. As can be seen, due to the higher friability of HPC samples compared to NC samples, more cracks appeared at higher loading levels.

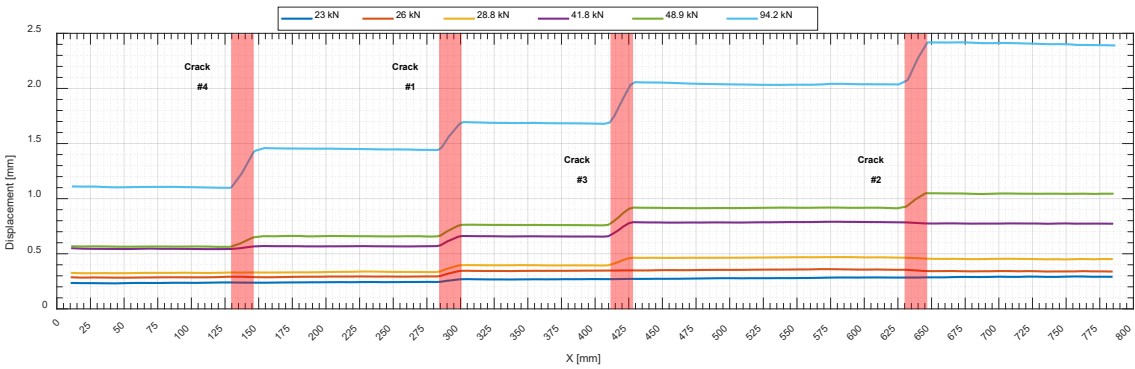

**Figure 10.** Surface displacement and order of crack appearance for specimen 2N100 under different loading levels.

**Table 4.** Cracking loads registered in the data acquisition system for all specimens.

| | ID | 1st Crack | 2nd Crack | 3rd Crack | 4th Crack | 5th Crack | 6th Crack | 7th Crack | 8th Crack |
|---|---|---|---|---|---|---|---|---|---|
| Normal Concrete | 1N150 | 43.6 | 50.7 | - | - | - | - | - | - |
| | 2N100 | 23.0 | 26.0 | 28.8 | 41.8 | 48.9 | - | - | - |
| | 3N150 | 49.9 | - | - | - | - | - | - | - |
| | 4N150 | 38.2 | - | - | - | - | - | - | - |
| | 5N100 | 25.7 | 27.4 | 33.7 | 44.7 | - | - | - | - |
| | 6N100 | 19.6 | 20.9 | 25.5 | 29.4 | - | - | - | - |
| | 7N100 | 23.5 | 25.2 | 41.1 | - | - | - | - | - |
| HPC | 8U100 | 6.8 | 21.6 | 26.8 | 31.0 | 33.9 | 88.5 | - | - |
| | 9U100 | 21.5 | 21.5 | 33.1 | 33.1 | 35.4 | 40.9 | 43.1 | 73.5 |
| | 10U100 | 20.7 | 26.1 | 26.1 | 33.9 | 31.1 | 38.5 | - | - |

Figure 11 shows the surface displacement and location of jumps as detected cracks (red boxes), together with strains monitored by the SGs installed on the embedded reinforcement for 2N100. As follows, debonding (failure) length, $L_{debonding}$, around the detected cracks was estimated by Equation (4). As can be seen, the estimated debonding zones around the cracks (orange boxes) developed with increasing loading level and affected the strain on rebar by cutting off concrete collaboration in those areas.

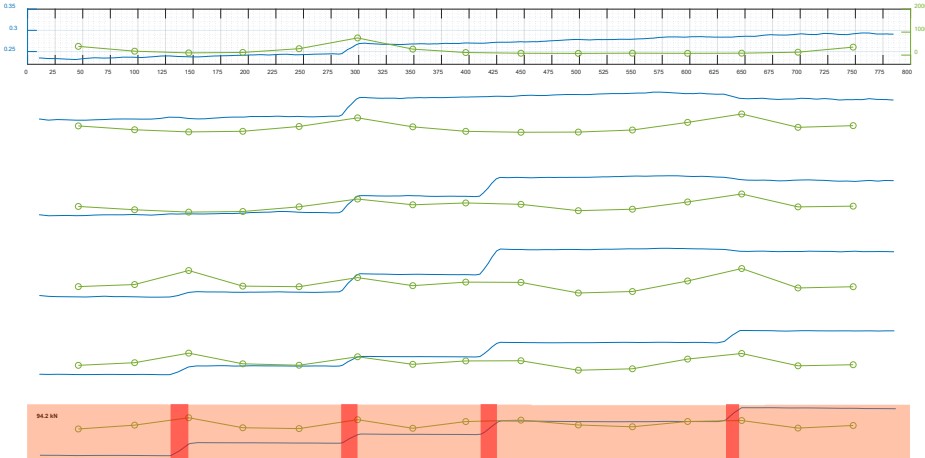

**Figure 11.** Surface displacement, debonding zones around detected cracks, and strain on embedded reinforcement under different loading levels.

Figure 12 illustrates the strains on the embedded reinforcement monitored by the SGs, those predicted by the proposed equation [5], together with surface deformations monitored by DIC on 2N100 under different loading levels.

After data collection was finished, the collected datasets from DIC, the installed SGs, and the proposed equation were arranged to generate the training dataset. Since the input dataset plays a vital role in developing a well-trained model, special care has been taken when generating the input datasets. For this purpose, two different groups of training datasets were generated, comprising: (a) data collected from only the installed strain gauges, called the experimental dataset, (b) combination of data collected from the strain gauges and generated synthetic data in debonding zones, called the hybrid dataset. Therefore, in the length of failure and slip zones, the synthetic dataset was calculated and combined with data collected from the strain gauges for the rest of the areas. After deriving both the experimental and hybrid training datasets over the RC members, the desired datasets were obtained to train the models with a hybrid learning approach.

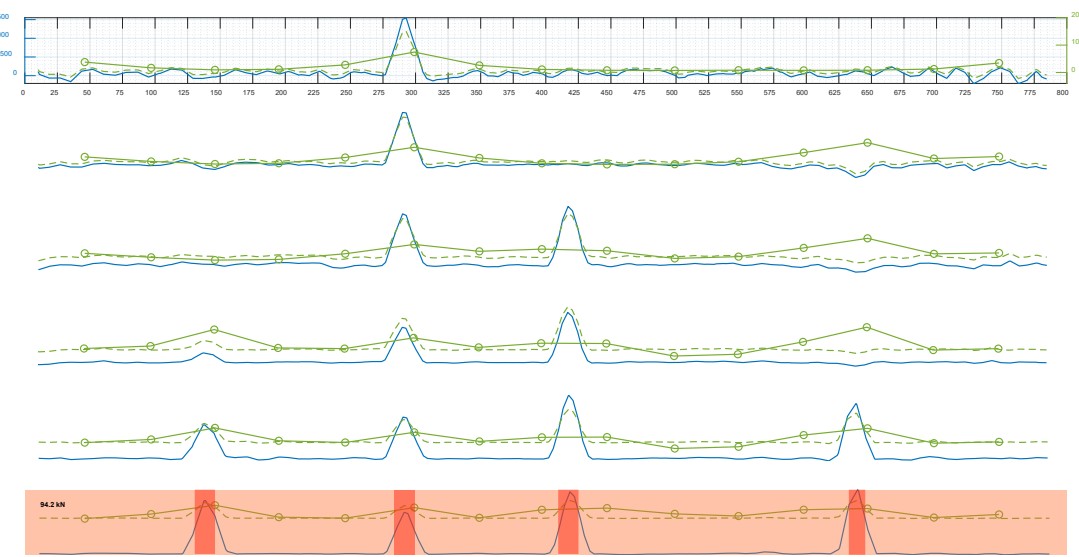

**Figure 12.** Surface deformation in cracking loads for 2-N100, together with strain in the embedded rebar and that predicted by the proposed equation.

### 4. Training ML Regression Models with Hybrid Learning Approach

Machine learning regression is a versatile tool for uncovering complex correlations between obvious and embedded data; however, generated datasets play a key role in developing regression models. In this study, the parameters that served as the input dataset were: surface deformations monitored by DIC, the reinforcement ratio $\rho = {}^{A_s}/_{A_c}$, $\eta = {}^{E_s}/_{E_c}$, and $f_{ct}$ as the tensile strength of concrete. The output was the response for local strain on the embedded rebar.

Four machine learning regression algorithms, including neural network (NN), Gaussian process regression (GPR), decision tree, and ensemble model, were used to generate regression models and predict strains on embedded reinforcement. Before training the models, the prepared datasets were divided into training and test groups at an 80/20 ratio to find the most efficient algorithm. It needs to be mentioned that the whole dataset collected from 2N100 was not observed in the training phase and was kept with test dataset to evaluate the performance of the model trained by the hybrid learning approach. Finally, the prepared dataset was pre-processed with data normalization along with shuffling to avoid bias towards a specific dataset.

#### 4.1. Dataset Normalization

Normalizing the prepared dataset to zero mean and unit variance allowed all input dimensions to be treated equally and facilitated better convergence. In this study, the range of the prepared dataset was drastically different, e.g., surface deformation was normally at the micro strain level, the reinforcement ratio was in percent scale, and tensile strength was in tenth scale. Therefore, data normalization was necessary to avoid bias towards a specific input variable, which ultimately affects the results. However, after solving the problem, the obtained results were again converted to achieve the actual data. Thus, the following normalization technique was used to keep all the parameters between 0 and 1.

$$Variable_i^{Normalized} = \frac{Variable_i - min(Variable_1, Variable_2, \ldots, Variable_i)}{max(Variable_1, Variable_2, \ldots, Variable_i) - min(Variable_1, Variable_2, \ldots, Variable_i)}, \tag{7}$$

$Variable_i$ is any original input parameter; $min(Variable_1, Variable_2, \ldots, Variable_i)$ is the minimum value of the similar parameter; $max(Variable_1, Variable_2, \ldots, Variable_i)$ is

the maximum value of the parameter; and $Variable_i^{Normalized}$ is the normalized value of the parameter.

*4.2. Verification of Trained Models, Hyperparameter Optimization, and Statistical Performance Measures*

Machine learning models feature several hyperparameters that can be tweaked to alter the algorithm's performance. Thus, the accuracy of training models mainly depends on how well the hyperparameters have been tuned. However, even with the same ML model, one combination of hyperparameters is not always the best for different training datasets.

**Neural networks (NN)** consist of fully connected layers, including hyperparameters such as the number of layers, size of layers, and activation function. Therefore, the application of NN in data regression analysis with different preset hyperparameters approaches the lowest feasible values of the root-mean-square error (RMSE) by optimizing the model. The defined loss function was the mean square error (MSE), as in most regression models in the literature review, and the "Adam" optimizer was used for "gradient descent backpropagation" with a learning rate of 0.0001. Then, the designed models were trained for 1000 iterations. Table 5 illustrates the RMSE values corresponding to different NN models to find the lowest obtained RMSE in both the validation and test datasets generated by the SGs andhybrid approach. For the experimental training dataset, an NN with three fully connected layers, ten nodes in each, and "tanh" as the activation function was found as the optimized model. In addition, for the hybrid training dataset, an NN with three fully connected layers with 100 nodes in each and "ReLU" as the activation function was found as the optimized model.

**Table 5.** Obtained RMSE values corresponding to different trained NN models with both experimental and hybrid datasets.

| Hyperparameters | | | RMSE | | | |
| --- | --- | --- | --- | --- | --- | --- |
| | | | SG Dataset | | Hybrid Dataset | |
| No. layers | Size of layers | Activation function | Validation | Test | Validation | Test |
| 2 | 10/10 | ReLU | 0.12103 | 0.12819 | 0.05591 | 0.055688 |
| | 10/10 | sigmoid | 0.12307 | 0.14326 | 0.05647 | 0.05614 |
| | 10/10 | tanh | 0.11657 | 0.13889 | 0.055903 | 0.05558 |
| | 100/100 | ReLU | 0.13841 | 0.14727 | 0.055305 | 0.053617 |
| | 100/100 | sigmoid | 0.12314 | 0.13028 | 0.058698 | 0.058416 |
| | 100/100 | tanh | 0.11951 | 0.13182 | 0.056387 | 0.055251 |
| 3 | 10/10/10 | ReLU | 0.11806 | 0.1224 | 0.055578 | 0.055638 |
| | 10/10/10 | sigmoid | 0.11567 | 0.13109 | 0.056542 | 0.056549 |
| | 10/10/10 | tanh | 0.1125 | 0.12234 | 0.055978 | 0.055739 |
| | 100/100/100 | ReLU | 0.13675 | 0.13626 | 0.054921 | 0.053619 |
| | 100/100/100 | sigmoid | 0.12083 | 0.12517 | 0.058223 | 0.056089 |
| | 100/100/100 | tanh | 0.12193 | 0.12869 | 0.056161 | 0.055888 |

Searching for the best combination of hyperparameters is time-consuming and tedious. That is why, regarding the state-of-the-art technique, only NNs with two and three fully connected layers were studied to find the optimized hyperparameters. For the other three algorithms, Bayesian optimization [16], which is a sequential design strategy for global optimization of black-box functions, was applied because it is more efficient.

**Decision Tree (DT)** is a hierarchical series of binary decisions in a tree-structured model. It contains three types of nodes, including root, interior, and leaf nodes, with hyperparameters including depth of tree and minimum leaf size (also called max leaf nodes). Minimum leaf size will allow the branches of a tree to have varying depths, which is a way to control the model's complexity. Therefore, to obtain optimized hyperparameters, the best minimum leaf size was searched using the Bayesian optimization method in the

range of 1 to $\frac{number\ of\ input\ data}{2}$, approaching minimum MSE. Figure 13 shows the minimum leaf size obtained using the Bayesian optimization method for both prepared datasets.

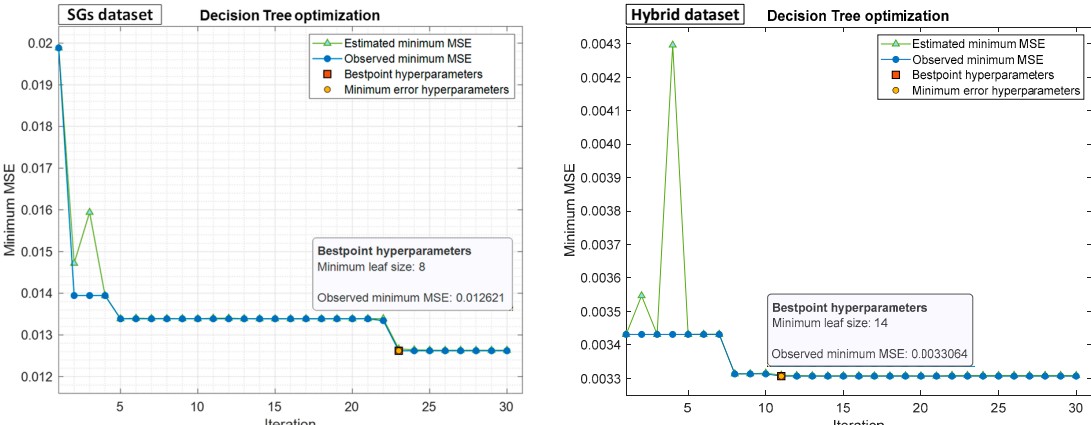

**Figure 13.** Minimum MSE plots for decision tree method in both experimental and hybrid datasets, which were updated by Bayesian optimization runs.

It needs to be mentioned that "observed minimum MSE" means the observed minimum MSE computed so far by the optimization process. "Estimated minimum MSE" corresponds to an estimate of the minimum MSE computed by the optimization process considering all the sets of hyperparameter values tried so far and including the current iteration. In addition, "best point hyperparameters" indicates the iteration corresponding to the optimized hyperparameters, and "minimum error hyperparameters" indicates the iteration corresponding to the hyperparameters that yield the observed minimum MSE.

**Gaussian Process Regression (GPR)** is a nonparametric Bayesian approach for regression analysis that creates a significant impression in machine learning. There are several benefits to GPR, including working well on small datasets and providing uncertainty measurements on predictions. Existing hyperparameters need to be optimized, including basis function, kernel function, and scale, together with the noise standard deviation used by the algorithm, called sigma. Hyperparameter optimization was performed in 30 iterations or less, searching ranges using the Bayesian optimization method for both the SG and hybrid datasets, as shown in Figure 14.

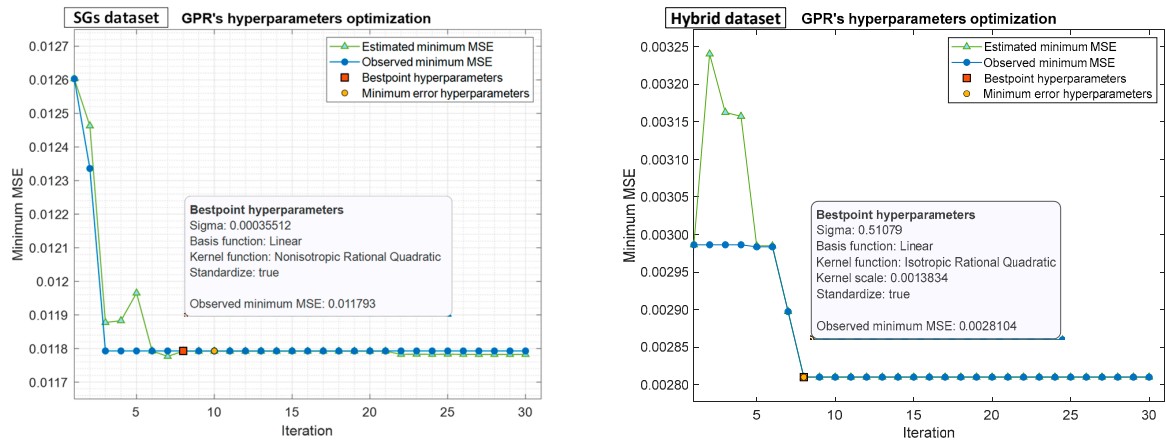

**Figure 14.** Minimum MSE plots for GPR method in both experimental and hybrid datasets, which were updated by Bayesian optimization runs.

**Sigma:** 0.0001–2.1309
**Basis function:** Constant, Zero, Linear

**Kernel function:** Nonisotropic Exponential, Nonisotropic Matern 3/2, Nonisotropic Matern 5/2, Nonisotropic Rational Quadratic, Nonisotropic Squared Exponential, Isotropic Exponential, Isotropic Matern 3/2, Isotropic Matern 5/2, Isotropic Rational Quadratic, Isotropic Squared Exponential

**Kernel scale:** 0.001731–1.731

**Ensemble model** is a technique that combines multiple models and then finds the best combination to produce improved results. For this aim, bootstrap aggregation (bagging) [17] and least-squares boosting [18] models were used to train the regression models. Then, Bayesian optimization was used to test the different combinations of hyperparameters for both the SG and hybrid datasets to find the optimized hyperparameters in the range of 10–500 learners, with a learning rate in the range of 0.001–1 and a minimum leaf size in the range of 1 to $\frac{number\ of\ observations}{2}$, as shown in Figure 15.

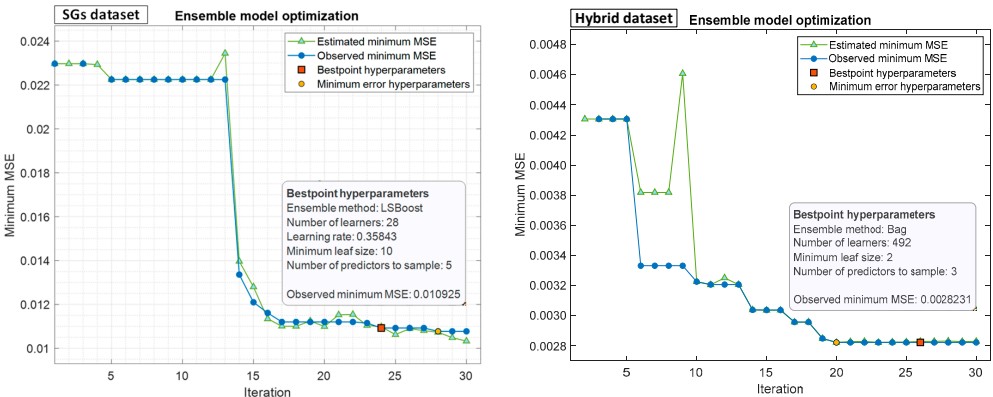

**Figure 15.** Minimum MSE plots for ensemble model method in both experimental and hybrid datasets, which were updated by Bayesian optimization runs.

The predictions by the four optimized models compared with true values in both the experimental (SG) and hybrid datasets, according to statistical parameters. Figure 16 shows the graphs comparing the true and predicted responses by the four optimized regression models, and Table 6 presents the summary of the validation and accuracy tests to find the most efficient models trained by each experimental and hybrid training dataset.

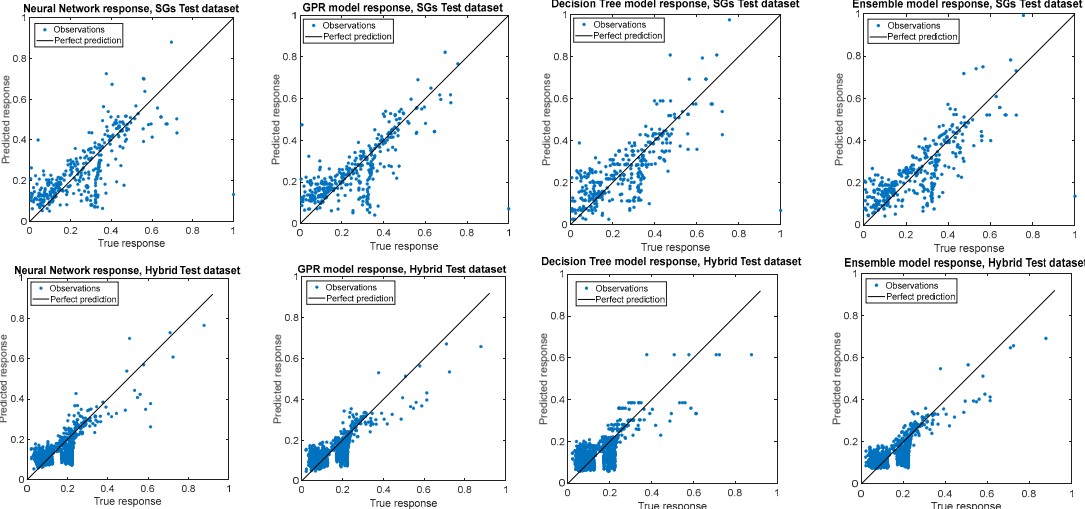

**Figure 16.** True and predicted responses by four optimized regression models trained by SG and hybrid datasets.

**Table 6.** Performance measures of each model considering both approaches in training and test phases.

| Models | | Neural Network | | GPR | | Decision Tree | | Ensemble | |
|---|---|---|---|---|---|---|---|---|---|
| Learning approach | | SG | Hybrid | SG | Hybrid | SG | Hybrid | SG | Hybrid |
| $RMSE = \sqrt{\frac{\sum_{i=1}^{n}\left(\Delta\varepsilon_{i,\,pre}-\Delta\varepsilon_{i,\,tru}\right)^{2}}{n}}$ | Validation | 0.120 | 0.055 | 0.108 | 0.053 | 0.112 | 0.057 | 0.104 | 0.053 |
| | Test | 0.125 | 0.054 | 0.116 | 0.051 | 0.119 | 0.059 | 0.112 | 0.052 |
| $R^{2} = 1 - \frac{\sum_{i=1}^{n}\left(\Delta\varepsilon_{i,\,tru}-\Delta\varepsilon_{i,\,pre}\right)^{2}}{\sum_{i=1}^{n}\left(\Delta\varepsilon_{i,\,tru}-mean(\Delta\varepsilon_{i,\,tru})\right)^{2}}$ | Validation | 0.680 | 0.510 | 0.740 | 0.550 | 0.720 | 0.470 | 0.760 | 0.540 |
| | Test | 0.530 | 0.570 | 0.600 | 0.600 | 0.570 | 0.480 | 0.620 | 0.590 |
| $MSE = \frac{\sum_{i=1}^{n}\left(\Delta\varepsilon_{i,\,pre}-\Delta\varepsilon_{i,\,tru}\right)^{2}}{n}$ | Validation | 0.014 | 0.003 | 0.012 | 0.003 | 0.013 | 0.003 | 0.011 | 0.003 |
| | Test | 0.015 | 0.003 | 0.013 | 0.003 | 0.014 | 0.003 | 0.012 | 0.003 |
| $MAE = \frac{\sum_{i=1}^{n}\left|\Delta\varepsilon_{i,\,pre}-\Delta\varepsilon_{i,\,tru}\right|}{n}$ | Validation | 0.088 | 0.042 | 0.078 | 0.040 | 0.080 | 0.043 | 0.076 | 0.041 |
| | Test | 0.088 | 0.041 | 0.076 | 0.039 | 0.081 | 0.044 | 0.078 | 0.041 |

As presented in Table 6, the ensemble model and GPR showed better performance on the training datasets generated experimentally (SG) and by the hybrid learning approach, respectively. To compare the performance of these two ML regression approaches, the residuals in normalized strain predictions are illustrated in Figure 17. Residuals represent the difference between any data point and the regression line, which can be called "errors," and are expressed as the difference between the predicted and observed values.

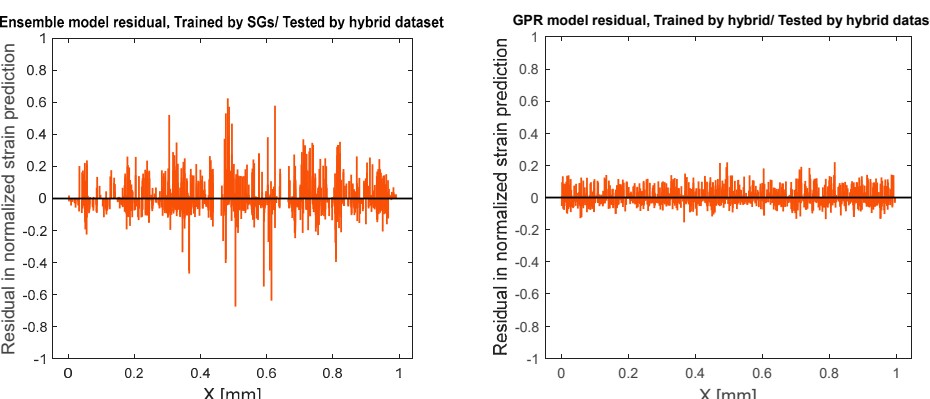

**Figure 17.** Obtained residuals in normalized strain predictions in the hybrid test dataset.

The role of concrete type and cross-sectional area in the predicted results was studied using the optimized models. Figure 18 shows the box plots providing a visualization of residuals in normalized predictions by both optimized models for each concrete type and cross-sectional areas. The bottom and top of each box are the 25th and 75th percentiles of the predictions, respectively. The distance between the bottom and top of each box is the interquartile distribution range (IQR), which is the spread of the middle 50% of the data values. The line in the middle of each box is the prediction median; in this case it was under zero, which showed that the predictions were mostly underestimated. The whiskers are lines extending above and below each box from the end of the interquartile range to the furthest observation within the whisker length, which is equal to 1.5*IQR. Observations beyond the whisker length were marked as outliers.

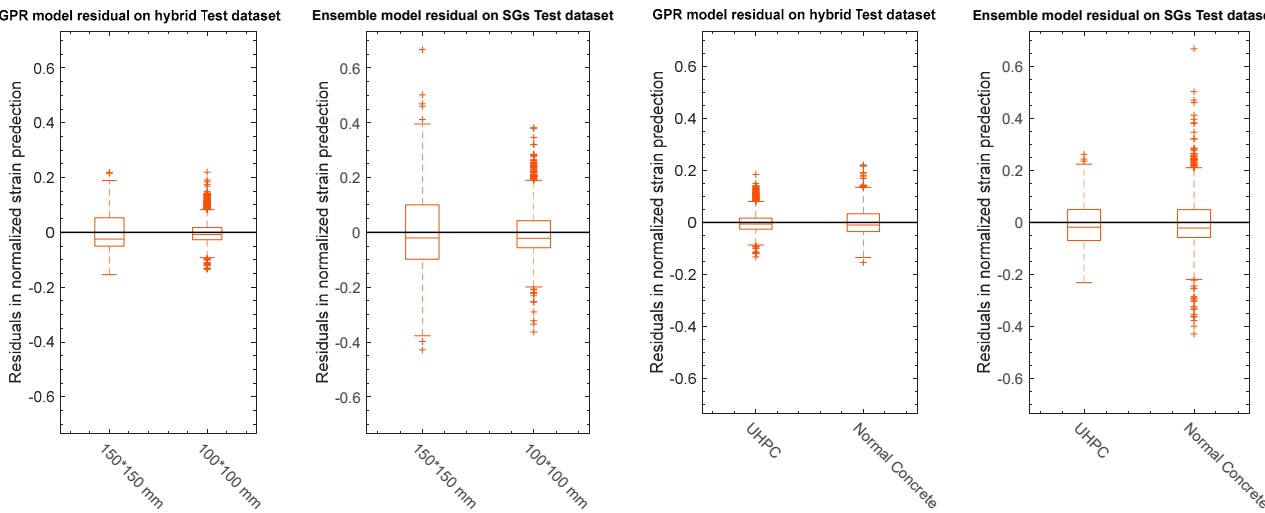

**Figure 18.** Box plot visualization of summary statistics for residuals in normalized predictions for each concrete type and cross-sectional area.

Overall, the GPR model trained with a hybrid learning approach showed the best performance among all the studied models. Figure 19 shows the predicted strain by the GPR model using a hybrid learning approach on embedded reinforcement for 2N100 under different loading levels. As a reminder, the obtained datasetfrom this specimen was not observed in the training phase and the predicted strain presented alongside the data collected from the installed SGs was used to verify the performance of the proposed method. There was obviously some outlier noise in the predicted values, which could be eliminated to improve the results by post-processing the predictions.

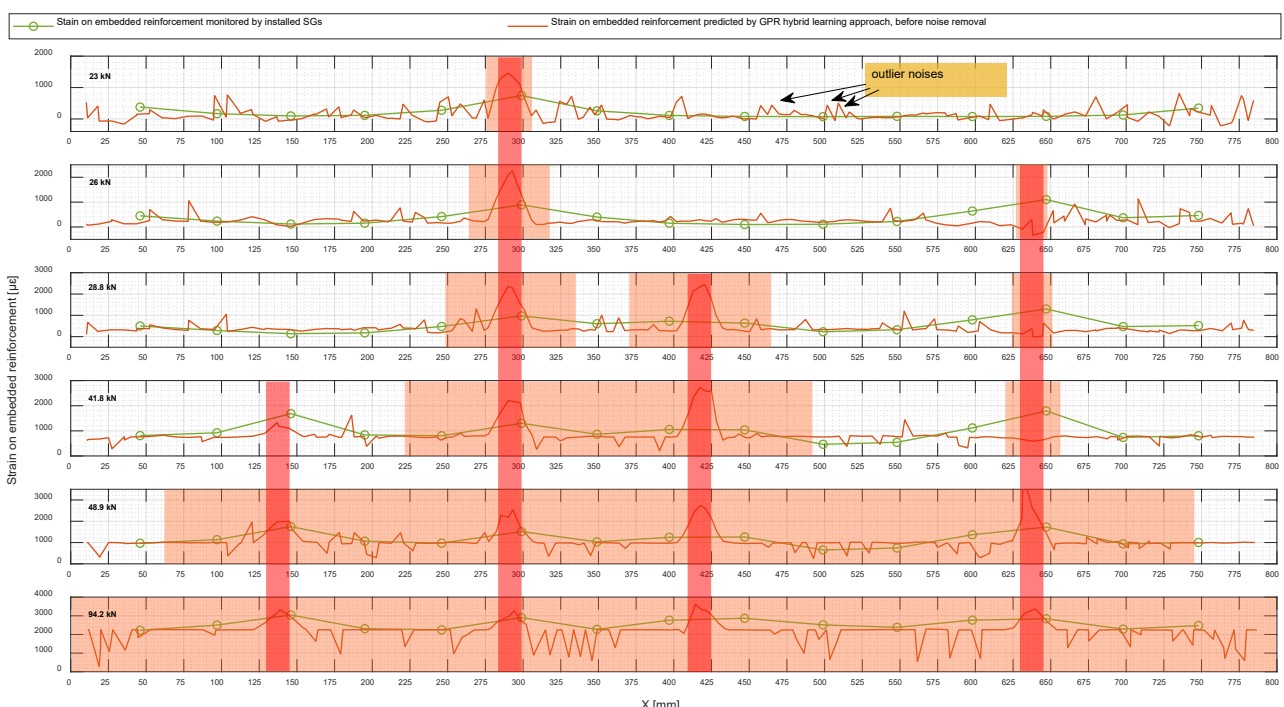

**Figure 19.** Predicted strain by hybrid learning approach on embedded reinforcement for 2N100 alongside that obtained by monitoring the installed SGs.

### 4.3. Data Improvement and Outlier Removal Using Hampel Identifier

The Hampel identifier is a statistical method used to remove outlier noise and improve the robustness of the obtained vector of prediction. The task was to perform noise removal on the predicted strain, represented by the strain ($\varepsilon$) as a sampled value along with the spatial location ($X$) on the defined domain of the specimen. Of the total number of dataset points considered "$n$", the spatial location and sampled value at the $j_n$ data node are given by $X_{j_n}$ and $\varepsilon_{j_n}$, respectively.

In spatial location ($X$), in a sequence of $X_1, X_2, X_3, \ldots , X_{j_n}$, a one-dimensional kernel with length of $(2d+1)$ is defined as a sliding window. Then, Equations (8) and (9) are used to calculate the point-to-point median ($m_i$) and standard deviation ($\sigma_i$) in six surrounding samples, with three samples ($d = 3$) in each side.

$$median\ in\ X\ :\ m_i{}^X = median\left( \varepsilon^{j_{i-d}}, \varepsilon^{j_{i-d+1}}, \varepsilon^{j_{i-d+2}}, \ldots , \varepsilon^{j_i}, \ldots , \varepsilon^{j_{i+d-2}}, \varepsilon^{j_{i+d-1}}, \varepsilon^{j_{i+d}} \right) \tag{8}$$

$$Standard\ deviation\ in\ X\ direction\ :\ \sigma_i{}^X = k\ median\left( \left| \varepsilon^{j_{i-d}} - m_i \right|, \ldots , \left| \varepsilon^{j_i} - m_i \right|, \ldots , \left| \varepsilon^{j_{i+d}} - m_i \right| \right) \tag{9}$$

Therefore, outlier noise can be identified when the difference between the sampled value and local median is higher than $t\sigma_i{}^X$, $t = 3$ [19]; then, replaced with the median, $\varepsilon_{j_n} = m_i{}^X$, as given by Equation (10):

$$\varepsilon_{j_n} = \begin{cases} \varepsilon_{j_n} & for\ clean\ data,\ \left| \varepsilon_{j_n} - m_i{}^X \right| \leq t\sigma_i{}^X \\ \varepsilon_{j_n} = m_i{}^X & for\ outlier\ noises,\ \left| \varepsilon_{j_n} - m_i{}^X \right| > t\sigma_i{}^X \end{cases} \tag{10}$$

Figure 20 shows the noise-removed prediction of strain in embedded reinforcement using the hybrid learning approach alongside that obtained by monitoring the installed SGs.

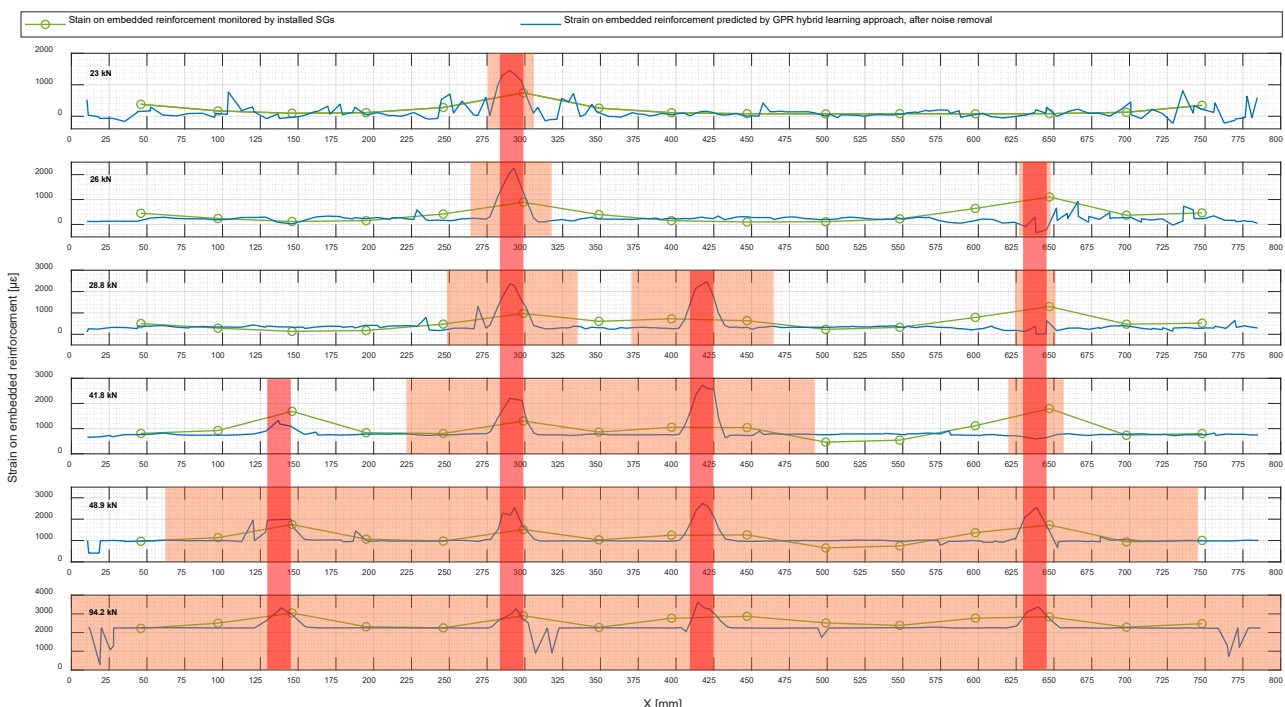

**Figure 20.** Noise-removed prediction of strain in embedded reinforcement alongside that obtained by monitoring the installed SGs.

## 5. Conclusions

The proposed work has established an innovative and efficacious machine learning (ML) regression technique with a hybrid learning approach to predict strain in embedded reinforcement. Application of ML regression can eliminate existing analytical modeling complexities in tensile RC members, which is caused by the tension stiffening effect. To generate a training dataset, ten specimens were prepared with different concrete types and cross-sectional areas and tested under pure tension with different loading patterns.

Due to the small amount of data obtained from the installed SGs, some synthetic data were additionally generated by a previously proposed semi-empirical equation. Therefore, the experimental and synthetic input datasets consisted of 384 and 1395 observations, respectively, generated with input variables comprising $\varepsilon_{LC}$, $F$, reinforcement ratio $\rho = {}^{A_s}/_{A_c}$, $\eta = {}^{E_s}/_{E_c}$, and $f_{ct}$, together with $\varepsilon_{LS}$ as the target value to train the model. To approach hybrid learning, both groups of data generated experimentally and synthetically were combined to enrich the training dataset. Then, the hyperparameters belonging to four different ML regression methods, comprising neural network (NN), Gaussian process regression (GPR), decision tree, and ensemble model, were optimized by the Bayesian optimization technique. Finally, the performance of the discussed models, which were first trained by only the experimental dataset and then by the hybrid dataset, were appraised through several statistical performance metrics. Notably, 20% of the whole dataset, including the data collected from 2N100, was considered as the test dataset. The prepared test dataset was not observed in the training phase and was used to check the performance of the trained models.

The evaluation of the proposed models was based on three different approaches. The first approach was finding the optimized regression model showing the best performance in strain prediction for each of the experimental and hybrid training datasets. For the experimental training dataset, the optimized ensemble model presented slightly better performance than the other studied models, which were all optimized as well. Regarding the hybrid training dataset, GPR showed better performance than the other studied regression models. All of the obtained statistical performance metrics were summarized in Table 6.

The second approach was assessing the performance of hybrid learning versus that of models trained by only the experimental dataset. As presented in Table 6, the RMSE achieved by the hybrid learning approach was roughly half of that of the models trained by only the experimental dataset. Hence, an exceptional improvement was attained. In addition, residual graphs for strain predictions were generated to assess the different methods and variables, including (1) ML regression using the experimental input dataset versus the hybrid learning approach, (2) different reinforcement ratios, and (3) different concrete types, and the following results were obtained:

1.  Residuals from the hybrid learning approach were lower than those from algorithms trained by only the experimental dataset.
2.  Lower reinforcement ratio, which means higher concrete cover, can increase the residual in strain prediction.
3.  Concrete type did not have much effect on ML regression performance, as long as the collected training dataset was sufficient to optimize the models.

In the third approach, 2N100 was implemented as a test specimen to observe the obtained predictions alongside strains monitored by the pre-installed strain gauges. It needs to be mentioned that the dataset collected from this specimen was not observed beforehand in the training phase. The model trained by the hybrid learning approach showed good performance after noise removal. Therefore, herein, the proposed model ensures a successful application for non-contact strain measurement in embedded reinforcement based on surface deformations.

In conclusion, this study successfully proposed and implemented a novel machine learning regression technique with a hybrid learning approach for predicting strain in embedded reinforcement. The results of the experiment showed exceptional improvement

compared to previous studies and demonstrated the effectiveness of the proposed method in non-contact prediction of strains on embedded rebar based on monitored surface deformations. This research provides valuable insight for a solution for the construction industry, and further studies need to be conducted to expand its applications in real-scale structures and validate its reliability.

**Author Contributions:** Conceptualization, A.M. and C.P.; Data Collection, C.P. and B.T.; Methodology and Software, A.M.; Machine Learning regression models, A.M.; Analysis, A.M., C.P. and B.T.; Writing original draft, A.M.; Writing—review and editing, A.M., C.P. and B.T.; Funding acquisition, C.P.; All authors have read and agreed to the published version of the manuscript.

**Funding:** This research was funded by FORMAS, project number 2019-01515. Any opinions, findings, and conclusions expressed in this paper are those of the authors and do not necessarily reflect the views of FORMAS.

**Data Availability Statement:** The data presented in this study are available on reasonable request from the corresponding author. The data are not publicly available due to further ongoing studies.

**Conflicts of Interest:** The funders had no role in the design of the study; in the collection, analyses, or interpretation of data; in the writing of the manuscript, or in the decision to publish the results.

## Abbreviations

| | |
|---|---|
| RC | Reinforced concrete |
| NC | Normal Concrete |
| SG | Strain Gauge |
| HPC | High-Performance Concrete |
| FOS | Fiber Optic Sensor |
| NN | Neural Network |
| DIC | Digital Image Correlation |
| DT | Decision Tree |
| ML | Machine Learning |
| GPR | Gaussian Process Regression |
| SHM | Structural Health Monitoring |
| RMSE | Root-Mean-Square Error |

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
