# Peer review of "Prediction of Strain in Embedded Rebars for RC Member, Application of Hybrid Learning Approach"

_infrastructures, doi:10.3390/infrastructures8040071_

Round 1
Reviewer 1 Report
The article is well-structured and well-discussed, but there are a few writing errors:
1. On the first page, HHPC is the wrong abbreviation, please correct it to UHPC.
2. In line 264, please add the correct heading for Table 4.
3. Most authors/publications define ultra high performance concrete (UHPC) as having a minimum strength in compression of 150MPa. If less conservative, this limit is defined by other authors as 120MPa. However, in your article (Table 2) the UHPC specimens are below these limits. How/why did you classify it as UHPC and not HPC.
Author Response
Response to Reviewer 1 Comments:
The article is well-structured and well-discussed,
Response 1: The authors would like to thank the reviewer for the acknowledgement of the manuscript.
but there are a few writing errors:
- On the first page, HHPC is the wrong abbreviation, please correct it to UHPC.
- In line 264, please add the correct heading for Table 4.
Response 2: Thanks for the comments. These parts are corrected.
- Most authors/publications define ultra high-performance concrete (UHPC) as having a minimum strength in compression of 150MPa. If less conservative, this limit is defined by other authors as 120MPa. However, in your article (Table 2) the UHPC specimens are below these limits. How/why did you classify it as UHPC and not HPC.
Response 3: We greatly appreciate the reviewer’s efforts to carefully review the paper. The authors are completely agree with the reviewer in this case. Therefore, UHPC is changed to HPC through all the manuscript.
Thank you again for your time and effort, and for helping us to improve the manuscript.
Reviewer 2 Report
Full title: Prediction of strain in embedded rebars for RC member, application of hybrid learning approach
I am very surprised with this work. This manuscript is having no novelty. The procedure described here can be found in youtube or in any tutorial in the web. Besides, there is no consistent results.
This is not research. The lack of results, scientific soundness or even a new methodology in present in this work.
The authors should explain the load-carrying mechanism of the embedded re-bars: how the load is transferred?? how it is carried by this material?? and more importantly how this material responds to it??
This article is more like an engineering report. In introduction, it is very obscure to understand the purpose of the paper, the authors did not clearly descript what was the purpose of the research? And the layout of the paper is not good that it was difficult to capture the core idea of the research. If you want to publish the content of the paper, please significantly revise the paper totally.
This reviewer recommends rejecting this paper and suggests the authors rewrite for brevity, understandability, and to highlight the unique contributions of the work, then resubmit.
Author Response
Response to Reviewer 2 Comments:
I am very surprised with this work. This manuscript is having no novelty. The procedure described here can be found in youtube or in any tutorial in the web. Besides, there is no consistent results.
This is not research. The lack of results, scientific soundness or even a new methodology in present in this work.
Response 1: We are unclear on this reviewer's suggestion here. If the editor feels this is an important point, please clarify this and we will attempt to address this.
The authors should explain the load-carrying mechanism of the embedded re-bars: how the load is transferred?? how it is carried by this material?? and more importantly how this material responds to it??
Response 2: Load-carrying mechanism of the embedded rebar is explained in Figure 2, and Figure 3. As you can see, Figure 3 shows how the tensile load is transferred along the rebar and the contribution of concrete in tension is illustrated in detail.
This article is more like an engineering report. In introduction, it is very obscure to understand the purpose of the paper, the authors did not clearly descript what was the purpose of the research? And the layout of the paper is not good that it was difficult to capture the core idea of the research. If you want to publish the content of the paper, please significantly revise the paper totally.
Response 3: We are unclear on this reviewer's suggestion here. If the editor feels this is an important point, please clarify this and we will attempt to address this.
Thank you for your time and effort, and for helping us to improve the manuscript.

Reviewer 3 Report
Kindly find the attached comments.

Author Response
Response to Reviewer 3 Comments:
In the reviewer’s opinion, this is a well-written paper, and the topic is generally relevant to the scope of MPDI Infrastructures.
Response 1: The authors would like to thank the reviewer for the acknowledgment of the manuscript.
however, it is recommended that the draft is carefully revised according to the following a few minor comments before considering it for publication.
Minor comments:
- Figure 12 displays the predicted strain obtained from the proposed equation, which exhibits a varied trend based on the loading magnitudes. Specifically, when the loading magnitude is 23kN, the predicted strain closely aligns with the strain monitored by Digital Image Correlaon (DIC). Conversely, under higher loadings such as 94.2kN, the predicted strain is more in line with the strain monitored by Strain Gauges (SGs). Please address this tendency of bias.
Response 2: Before appearance the first crack, reinforced concrete shows elastic behavior, and there is fully bonded between concrete and rebar. Therefore, strain in embedded rebar is super close to observed strain on the surface of specimens. In other hand, by increasing tensile stress, due to development of debonding zones and reduction of concrete collaboration in transferring tensions, strain in embedded rebar is higher than that of observed on the surface.
- Use scientific notation for the following:
- Line 222: 7,14 pixels -> 7.14 pixels
- Line 223: 59,0% -> 59.0%
- Line 248: mm2 -> ??2
- The table in line 264 should be Table 4. Also, the table title must be corrected.
Response 3: We greatly appreciate the reviewer’s efforts to carefully review the paper. The typing errors were corrected.
- An additional explanation for the Table 4 must be provided.
Response 4: The authors are completely agreed with the reviewer in this case. Then, additional information is added and the paragraph in page 9, lines 258-263, is edited.
Thank you again for your time and effort, and for helping us to improve the manuscript.